# The Potential of *Lactiplantibacillus plantarum* ATCC 14917 in the Development of Alginate-Based Gel Formulations with Anti–*Staphylococcus aureus* Properties

**DOI:** 10.3390/ph16081112

**Published:** 2023-08-07

**Authors:** Monaliza Teresa Campos Sodré, Fernanda Avelino Ferraz, Amanda Karoline Vieira Alencar, Karinny Farias Silva, Douglas Henrique dos Santos Silva, Lucas dos Santos Silva, Jéssica Silva dos Santos Araújo Carneiro, Cristina Andrade Monteiro, Luis Cláudio Nascimento Silva, Andrea de Souza Monteiro

**Affiliations:** 1Laboratory of Applied Microbiology, CEUMA University, São Luís 65075-120, MA, Brazil; 2Laboratory of Microbial Pathogenesis, CEUMA University, São Luís 65075-120, MA, Brazil; 3Laboratory of Microbiology Research, Federal Institute of Education, Science and Technology of Maranhão (IFMA), São Luís 65030-005, MA, Brazil

**Keywords:** *Staphylococcus aureus*, skin infections, topical agents, probiotics, postbiotics

## Abstract

This study aimed to evaluate the potential of lactic acid bacteria (LAB) in developing alginate-based gel formulations to inhibit *Staphylococcus aureus*. Initially, the antagonistic actions of three lactic acid bacteria (LAB) (*Lacticaseibacillus rhamnosus* ATCC 10863, *Lactiplantibacillus plantarum* ATCC 14917, *Limosilactobacillus fermentum* ATCC 23271) were evaluated against *S. aureus* ATCC 25923. All tested LAB inhibited *S. aureus*, but the highest activity was observed for *L. plantarum* ATCC 14917 (*p* < 0.05). The antimicrobial effects of *L. plantarum* ATCC 14917 cell suspensions, sonicate cells extract, and cell-free supernatants (pH 5 or 7) were analyzed using a broth-based assay. The cell suspensions inhibited *S. aureus* at concentrations ≥ 10%, and these effects were confirmed by a time-kill assay. Alginate-based gels were formulated with cell suspensions, sonicate cells extract, and cell-free supernatant (pH 5). These formulations inhibited *S. aureus* growth. Based on the results, the alginate gel with cell suspensions at 10% was selected for further characterization. *L. plantarum* ATCC 14917 survived in the alginate-based gel, especially when stored at 5 °C. At this temperature, the *L. plantarum*-containing alginate gel was stable, and it was in compliance with microbiological standards. These findings suggest it can be a promising agent for the topical treatment of infections induced by *S. aureus*.

## 1. Introduction

The skin is considered the largest organ of the human body and the first biologically active immunological barrier, inhabited by diverse microorganisms in different areas of the body, such as bacteria, fungi, and viruses, acting as constituents in the skin protection barrier that forms the skin microbiota [1]. This microbiota is a community of commensal bacteria, one of the leading agents in skin defense mechanisms and immunological responses [2].

Specific skin characteristics, such as humidity, temperature, hydrogen potential (pH), and sebaceous secretion, among others, are intrinsically linked to the constitution of the microbiome [3,4,5]. Imbalances in the constitution of the skin microbiota lead to disturbances in the homeostasis of the skin, allowing growth and colonization by opportunistic pathogens, resulting in dermatological diseases such as atopic dermatitis, psoriasis, and acne [3,4,5].

*Staphylococcus aureus* is one example of a pathogenic species able to induce skin diseases, such as atopic dermatitis (AD), also known as atopic eczema. AD features include a reduced microbial variety that favors the microbial colonization of the skin, inducing cutaneous inflammatory pruritic eruptions and triggering the production of toxins due to disturbances in the barrier function [6,7]. AD is a multifactorial pathology that is more prevalent in children and affects about 10% of the adult population [8].

The topical and systemic uses of products containing probiotics and/or their cellular lysates (postbiotics) have become widespread in recent years because of their ability to modulate the microbiota and reestablish its balance in favor of health [9,10]. These products must have essential characteristics, such as the following: (i) the strain(s) should be characterized by genetic and/or phenotypic assays; (ii) the probiotics formulations should have live microorganisms (probiotics); and (iii) scientific evidence regarding the dosage, duration, and delivery method for receiving the product should be available [11]. Postbiotics are defined as the “preparation” of inanimate microorganisms and/or their components that benefit the health of the target host. Probiotic- and postbiotic-derived topical formulations are used for restructuring the beneficial functions of dermatological health and for consequently improving the microbiota of the skin [12,13].

The interest in innovative products that deliver topical substances to the skin is exponentially growing. These products should be friendly to the skin microbiota, contributing to its balance [14]. It is also expected that they offer functions through the release of their bioactive molecules, such as the ability to inhibit pathogenic agents of the skin and immunomodulation [15].

In recent studies, various lactic acid bacteria (LAB) strains have been highlighted because of their beneficial effects in treating inflammatory diseases, such as AD [16,17]. The LAB products are formulated using different matrices that should maintain their effectiveness, viability, and stability [18,19,20]. In this sense, the present work aimed to develop a topical formulation containing probiotics or postbiotics with antimicrobial effects against *S. aureus*. The viability and stability of the gel-based formulation were evaluated.

## 2. Results

### 2.1. Antimicrobial Effects of Selected LAB against Staphylococcus aureus

Initially, the antimicrobial effects of the selected LAB strains were evaluated using the spot overlay assay. All three strains showed high inhibition capacity with IZ > 6 mm (Figure 1). Among the strains, *L. plantarum* ATCC 14917 presented the best inhibitory effects (18.6 ± 0.1 mm; *p* < 0.0001), followed by *L. rhamnosus* ATCC 10863 (8.75 ± 0.3 mm), and *L. fermentum* ATCC 23271 (10.8 ± 0.2 mm)

Next, we verified the antimicrobial activity of *L. plantarum* ATCC 14917 using a suspension of cells, sonicated cells, and supernatants (pH 5 or pH 7) in a broth-based assay. The growth of *S. aureus* was inhibited by whole-cell suspension at concentrations from 30% to 50%. The supernatant that did not obtain its stabilized pH (5.0) also resulted in antimicrobial activity from 20% to 50%, both bacteriostatic at the mentioned concentrations. The sonicate and the supernatant with stabilized pH (7.0) did not show inhibition of the pathogenic microorganism at all tested concentrations (Table 1).

### 2.2. Time-Death Assay

The results of the time-death assay are shown in Figure 2. The cells suspensions of *L. plantarum* ATCC 14917 at concentrations ≥ 20% were capable of reducing the growth of *S. aureus* after 3 h of incubation, while for the cell suspensions at 5% and 10%, inhibition was observed at 8 h and 4 h, respectively. From then on, the CFU decreased over time, resulting in cell death of *S. aureus* in all concentrations, except the cell at 5% with only a reduction of 2 log CFU/mL in the final period of the test. At 50%, cellular inviability occurred after 12 h of testing. For the other concentrations, the *S. aureus* population was approximately 6 log CFU/mL after 24 h.

### 2.3. Antimicrobial Effects of Probiotic and Postbiotic Gels

The agar diffusion assay was applied to evaluate the antimicrobial effects of each formulation containing cells, sonicated cells, or supernatant (adjusted to pH 5) of *L. plantarum*. The highest IZs were observed for gels containing supernatant at 20% and 30%. Moderate IZs were observed for the formulations containing cells at 10%, 20%, and 30%. The formulations in which samples of sonicated bacteria were used did not show inhibition halos (Table 2).

### 2.4. Viablity of Lactiplantibacillus plantarum ATCC 14917 in the Formulation

The evaluation of *L. plantarum* ATCC 14917 viability in the gel was performed at different temperatures (37 ± 2 °C, 25 ± 2 °C, and 5 ± 2 °C) (Figure 3). The results indicated that the *L. plantarum* ATCC 14917 viability in the gel kept at 37 °C decreased by 4 log CFU/g on the 7th day, and no growth was detected on the 14th day. For the sample stored at 25 °C, the bacterial population was stable after 7 days and showed progressive reduction until the 60th day reaching about 5 log CFU/g. Regarding storage at 5 °C, the viability was maintained until the 14th day, and the bacterial population reached 7.3 log CFU/g on the 60th day.

### 2.5. Formulation Stability Tests

Initially, the sodium alginate gel containing 10% of *L. plantarum* was submitted to the centrifugation test, where the process was carried out with three cycles of 30 min at an rpm of 3500. The gel did not present changes in its organoleptic characteristics, coloring, odor, and phase precipitation and was thus approved to be used in subsequent stability tests.

#### 2.5.1. Preliminary Stability Test (PS)

Gel samples were analyzed for fourteen days in the preliminary stability test (PS), interspersed daily between two different temperatures: freezer 10 ± 2 °C and oven 45 ± 2 °C. During this period of monitoring, they did not show changes in organoleptic characteristics (appearance, color, and odor), remaining stable. Regarding the physical–chemical characteristics (density and pH), the gel showed considerable changes within the standard (Table 3).

#### 2.5.2. Accelerated Stability Test (AS)

After the results were obtained from the PS test, the organoleptic and physicochemical characteristics continued to be evaluated for a more extended period (60 days) in the AS test, except for spreadability, which was evaluated only in days T0 and T60 (Table 4). These characteristics were analyzed at 37 ± 2 °C, 25 ± 2 °C, and 5 ± 2 °C. During the analysis of the results over the proposed time, it was observed in the appearance, color, and odor that the gel stored in an oven showed changes from the first week (7 days). At room temperature, it showed mild changes in color and odor after 30 days of the handling process. In addition, at a cool temperature, it did not present organoleptic alterations. Regarding pH and density, an alternation of values is observed at all tested temperatures. Regarding spreadability, no significant changes were observed.

### 2.6. Microbiological Evaluation Test

The results obtained in the microbiological evaluation indicated values below the standard limits when the gel with LAB was stored at 5 °C during the test period. When stored at 37 °C, the result exceeded the CFU/g limits for total fungi/yeasts. (Table 5)

## 3. Discussion

In the present study, the potential of selected LAB strains for the development of topical formulation against *S. aureus* was evaluated. *S. aureus* is an important opportunistic pathogen of the skin and other tissues [21]. For instance, an increase in the proportion of *S. aureus* and a decrease in microbial diversity are observed in patients affected by AD [22,23].

Initially, the antagonistic effects of the three LAB strains against *S. aureus* were verified. All selected LAB showed antimicrobial activity against *S. aureus*, with higher effects observed for *L. plantarum* ATCC 14917. The antagonistic effects of LAB are related to the production of antimicrobial metabolites, such as bacteriocins and other biologically active compounds [24,25]. The inhibitory effect against *S. aureus* may also be related to the medium’s acidification and the capacity of LAB strains to produce and tolerate high concentrations of lactic acid, inducing competitive exclusion through nutrients and ecological niches. Furthermore, the compounds released by LAB strains can interfere with the quorum system, producing immunomodulatory effects and favoring the action of host immune cells [24,25,26].

The antimicrobial effects of LAB strains are well documented in several studies. For instance, some studies have shown that *L. plantarum* strains have antibacterial effects against Gram-positive and Gram-negative bacteria, including *S. aureus* [27,28,29]. These effects are due to its broad antibacterial spectrum, producing a variety of bacteriocins [29,30,31,32,33,34,35,36,37,38,39]. In our work, the sonicate *L. plantarum* ATCC 14917 cell extracts did not present antimicrobial activity, corroborating with other studies [40]. The most likely hypothesis for the activity loss is that the acidic medium can inactivate some substances and other inhibitory factors [41]. It is important to highlight that other studies have shown antimicrobial activity of postbiotic preparations [42,43,44,45].

Given the antimicrobial effects of LAB, the strains have been proposed for the development of topical formulations for the treatment of skin infections, such as AD. Indeed, some clinical trials showed the benefits of the use of probiotic-based formulations in AD patients [46,47]. Based on these, alginate-based probiotic and postbiotic gels were formulated using *L. plantarum* ATCC 14917. The anti-staphylococcal capacity of the gel with LAB was obtained with a cell concentration of 10% in the agar diffusion test, the lowest percentage in the time-kill curve that showed the death of *S. aureus.* It was observed that the probiotic formulations were more suitable for use. Importantly, *L. plantarum* ATCC 14917 showed viability in the alginate gels, especially when stored at 5 °C.

Next, the stability of gel containing *L. plantarum* ATCC 14917 was evaluated, and no significant changes were detected in its organoleptic and physical–chemical characteristics. The results corroborated with other studies that used alginate-based formulations to deliver LAB [46,47]. The alginate-based formulations provide protection, a controlled release of active principles, and better formulation stability [48,49,50,51,52,53].

Storage conditions can also influence the viability and stability of the probiotic gel, which may lose its viability and undergo organoleptic and physical–chemical changes in a few days or weeks. It was observed that the bacteria incorporated in the gel stored at 5 °C remained viable, and the gel was stable throughout the assay. The same happened in other studies, where the formulations showed better viability and stability when stored at 5 °C and 4 °C [54,55].

The storage temperature is fundamental to determining a longer or shorter stability time in the products [56]. Usually, temperatures above 25 °C have a greater propensity for physical–chemical alterations and/or possible microbiological contamination.

The maintenance of formulation stability and bacterial viability is essential to ensure the functionality and commercialization of a probiotic formulation. Changes in pH occur because of hydrolysis, impurities, decomposition due to storage time, and inadequate transport and storage conditions according to the product. A decrease in density may indicate the incorporation of air or loss of volatile ingredients.

## 4. Materials and Methods

### 4.1. Bacterial Strains

The bacterial strains used in the study belong to the culture bank of the Laboratory of Applied Microbiology at the CEUMA University, São Luís, Brazil: *Staphylococcus aureus* ATCC 25923, *Lacticaseibacillus rhamnosus* ATCC 10863, *Lactiplantibacillus plantarum* ATCC 14917, and *Limosilactobacillus fermentum* ATCC 23271. The LAB strains were previously cultivated in tubes of 15 mL broth of Man, Rogosa, and Sharpe (MRS; Himedia, Mumbai, India) for up to 48 h at 37 °C. Then, the cultures were centrifuged for 15 min at 3800 rpm, at 4 °C, storing the supernatant and the pellet for later tests.

### 4.2. Antagonism Test–Spot Overlay of LAB

Initially, the antagonistic actions of the LAB strains (*L. rhamnosus* ATCC 10863, *L. plantarum* ATCC 14917, *L. fermentum* ATCC 23271) against *S. aureus* ATCC 25923 were achieved using the spot overlay technique described by Chew et al., 2015 [57], with minor modifications. Recent cultures of each strain were used to prepare microbial suspensions with approximately 10^8^ CFU/mL. Afterward, 10 µL of each bacterial suspension was inoculated in the central part of the MRS agar plate for 48 h at 37 °C in aerobic conditions. Then, the plates were superimposed with 15 mL of Mannitol salt agar with the *S. aureus* ATCC 25923 inoculum pre-adjusted by the MacFalard turbidity scale of 0.5. The plates were then incubated for 24 h at 37 °C to observe the development of inhibition zones (IZs). The results were evaluated considering the Equation (1):IZ = (dnib − dspot)/2(1)
where dnib is the diameter of the inhibition zone around the “spot” and dspot is the diameter of the probiotic growth zone.

The interpretation of the results is as follows:No inhibition capacity: R < 2 mm;Low inhibition capacity: R = 2–5 mm;High inhibition capacity: R > 6 mm.

This test was performed in triplicate.

### 4.3. Preparation of Postbiotics of Lactiplantibacillus plantarum ATCC 14917

*L. plantarum* ATCC 14917, the strain with the highest antagonist potential, was selected for further assays. Before the assay, recent *L. plantarum* ATCC 14917 cultures were centrifugated at 3800 rpm for 15 min and washed twice with saline (0.9% NaCl). The supernatants (at pH 5 and 7) were filtered through a sterilizing 0.22 micrometer membrane. Subsequently, the cell pellets were separated from the supernatants and standardized in percentages.

The pellets were homogenized in phosphate-buffered saline (PBS; a pH of ~7.4) with a final volume of 6.5 mL. Subsequently, they were taken to the sonication stage for 10 cycles of 60 s pulse on with 20 s vibration off at a frequency of 90 Hz. The samples were placed on ice during the sonication process. The cell lysis was confirmed by plating the samples in MRS agar and using optical density measurement (OD_600_ nm).

### 4.4. Inhibitory Action in Broth Media

The obtained cells were sonicated to prepare the sonicated extract. Each type of postbiotics was diluted in 400 µL of a solution consisting of 50% Muller–Hinton broth and 50% MRS broth to obtain different concentrations (5, 10, 20, 30, 40, and 50%) in 96-well polystyrene plates. Afterward, 10 µL of the *S. aureus* ATCC 25923 inoculum (approximately 10^8^ CFU/mL) was added. Positive control wells contained Muller–Hilton broth medium plus 10 µL of pathogenic bacteria, and, as a negative control, only the medium corresponded to the test. The microplates were inoculated and incubated in aerobic conditions at 37 °C for 24 h. After the incubation period, the wells were checked visually for bacterial growth.

### 4.5. Time-Death Curve

The antimicrobial potential against *S. aureus* ATCC 25923 was further analyzed using the time-to-death assay, *L. plantarum* ATCC 14917, based on the method proposed by Kang et al. in 2017 with modifications [58]. The *S. aureus* ATCC 25923 inoculum was prepared in MRS, and MH medium (1:1) and adjusted to a density of 10^6^ cells/mL. The *L. plantarum* ATCC 14917 cells were added in tubes to obtain different concentrations (5, 10, 20, 30, 40, and 50%). The tubes were incubated at 37 °C in aerobiosis. Aliquots of the cell suspensions (100 µL) were removed at predetermined time points (6 h, 12 h, 18 h, 24 h, and 30 h) for CFU enumeration.

### 4.6. Preparation of Gel Formulation Containing Probiotics

Glycerin (Anhydrous ACS) was used as a moisturizing and humectant agent, whose function is to prevent the loss of hydration and retain it by attracting water molecules. The formulation was prepared using pure sodium alginate (Isofar, Duque de Caxias, Brazil). For the process of homogenization of the components of the formulation, distilled water autoclaved at 121 °C for 15 min was used after the sterilization process at room temperature, starting with the homogenization of the sodium alginate in an appropriate container, and soon after, it was homogenized in the vortex for 20 to 30 min. After this period, glycerin was added, homogenizing again in the vortex for up to 3 min. The process of weighing the components of the formulation was carried out on a precision analytical balance; soon after, it was fractionated and packed in transparent polyethylene pots with screw-type lids, performed in triplicate. The formulation manipulation process was performed in laminar flow and analyzed following the Brazilian Pharmacopeia (Table 6).

### 4.7. Antimicrobial Activity of The Formulation

The *S. aureus* ATCC 25923 suspension (0.1 mL at 10^8^ CFU/mL) was added to 10 mL molten Mueller–Hinton agar medium (Himedia) stabilized at 45 °C. The mixture was poured into 90 mm Petri dishes (20 mL of medium/plate). Once the medium was solid, 9 mm diameter wells were made with sterilized plastic straws and filled with the formulations containing whole LAB cells (5%, 10%, 20%, and 30%), sonicated LAB cells, or centrifuged supernatant. A topical formulation without the active ingredients was used as a negative control. Plates were incubated at 37 °C for 24 h. The interpretation of the results was carried out according to the IZ, where the microorganisms were classified as sensitive (IZ > 3 mm), moderately sensitive (2 mm > IZ > 3 mm), and resistant (IZ ≤ 2 mm).

### 4.8. Viability of Probiotics in The Formulation

The test aimed to analyze the viability of *L. plantarum* cells when introduced in the formulations, evaluating their survival in specific periods (T0 as control, T7, T14, T21, T30, T60 days). Samples of the products were manipulated and distributed in sterile 50 mL falcon tubes and later tested using the plating method and incubated for up to 48 h at 37 °C in aerobiosis to verify the CFUs in the determined periods.

### 4.9. Stability Tests

The physicochemical stability tests of the formulations were carried out according to the specifications of the Quality Control and Stability Guide for Cosmetic Products and Brazilian Pharmacopoeia 2nd edition [59].

#### 4.9.1. Preliminary Stability Test

##### Centrifugation Test

Initially, the centrifugation test was carried out, weighing about 5 g of the formulation into 15 mL falcons, carried out in triplicate, at room temperature (24 ± 2 °C) with a rotation speed of 3000 rpm for thirty minutes, which was repeated three times. This test aimed to evaluate any modification in the product’s appearance, such as precipitation, phase separation, the formation of compact sediment, and coalescence, among others. If there were no such changes, the preliminary tests were continued.

##### Thermal Stress Test

After performing and passing the centrifugation test, the samples were submitted and interspersed every two days at 10 ± 2 °C and 45 ± 2 °C and monitored for 14 days at times of T0, T2, T4, T6, T8, T10, T12, and T14. At each reading of the results, the samples were evaluated according to their organoleptic, physical–chemical characteristics: appearance, color, odor, pH, and density.

#### 4.9.2. Accelerated Stability Test (AS)

The formulation samples were stored under different temperature conditions—25 ± 2 °C, 5 ± 2 °C, and 37 ± 2 °C—for 60 days. The organoleptic, physical–chemical, and chemical alterations were evaluated, reading the analysis results at time T0 (control), T7, T14, T21, T30, and T60, where the samples were analyzed in triplicate at all the mentioned temperatures. Refrigerator samples were used as standard samples, with minor changes expected in the product.

##### Evaluated Parameters

Organoleptic Evaluations

A macroscopic evaluation was carried out regarding the organoleptic characteristics of the samples: appearance (phase separation, precipitation, and turbidity), color, and odor, verifying the state of the samples during the proposed period.

Aspect

A visual analysis of the sample characteristics was performed, where macroscopic changes were observed in the established standard sample. The sample was classified according to the following criteria:

N: normal, without change;

LS: slightly separated, slightly precipitated, or slightly cloudy;

S: separated, precipitated, or cloudy;

IS: Intensely separated, cloudy, or precipitated.

Color and Odor

An evaluation of the color of the samples was performed visually. Regarding the odor, the evaluation took place through olfactory analysis compared to the established standard sample. It was classified according to the following criteria:

N: normal, no change;

LM: slightly modified;

#### 4.9.3. Physical–Chemical Evaluations

##### pH Value

The pH value was verified using an ATC digital pH meter, initially calibrated with standard solutions between 4 and 7, and determined through a 10% aqueous dispersion (1:10) of the sample in distilled water at room temperature.

##### Density

An analysis was carried out for the apparent density according to ANVISA through the direct relationship between the mass of the product and the specific volume occupied, measured with the aid of a previously weighed graduated cylinder and an analytical balance. Its density in g/mL was verified.

The values obtained were introduced into the apparent density formula, where

D = apparent density in g/cm;

m = mass of the sample in g;

v = final volume in mL; and

D = *m*/*v*.

##### Spreadability

The spreadability test was conducted to verify the product’s ability to spread, simulating its use on the skin tissue. An amount of 1 g was placed on a glass plate on a graph paper scale, and a glass plate equal to approximately 200 g was placed on the sample. After 1 min, the covered surface was measured using a graduated ruler, measuring the diameter in two opposite positions, and the overall diameter was calculated using the average diameter formula. The spreadability (Ei), determined at 25 °C, was calculated using Equation (2):Ei = [(d2) . π] /4(2)
where Ei is the spreadability of the sample for mass i (mm); d is the mean diameter (mm); and π is 3.14

### 4.10. Microbiological Evaluation

The microbiological evaluation was performed using plate counts of total viable microorganisms (bacteria, fungi, and yeasts), as recommended in the Brazilian Pharmacopoeia, 2nd edition [23]. For the microbiological evaluation, a microbial load was allowed within the maximum limits for mesophilic bacteria of 10^3^ CFU/g fungi and yeasts of 10^2^ CFU/g or mL. Sabouraud dextrose agar and casein soy agar (TSA—Tryptic Soy Agar) were used for fungi and yeasts, and bacteria, respectively. The plates containing Sabouraud dextrose were incubated in an oven at 25 °C ± 2 °C, and the TSA plates were incubated in an oven at a temperature of 35 °C ± 2 °C, both for 72 h. At the end of the incubation period, colony counts were performed on the plates where growth occurred and were reported in CFU/g.

### 4.11. Statistical Analysis

The results were expressed as the mean ± standard deviation of three independent experiments performed in triplicate. The data were analyzed by one-way analysis of variance (ANOVA) or two-way ANOVA. *p* values *<* 0.05 were considered significant.

## 5. Conclusions

The results obtained in this study show that *L. plantarum* ATCC 14917 inhibits the growth of *S. aureus* and is suitable for the development of topical formulation against this pathogen. The probiotic was able to survive in the alginate-based gel, especially when stored at 5 °C. At this temperature, the *L. plantarum*-containing alginate gel was stable and in compliance with microbiological standards. Taken together, these findings suggest it can be a promising agent for the topical treatment of infections induced by *S. aureus*.

## Figures and Tables

**Figure 1 pharmaceuticals-16-01112-f001:**
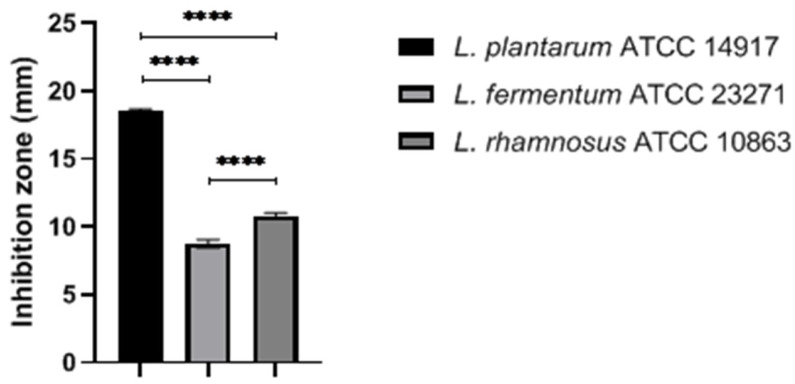
Antagonistic activity of selected LAB against *Staphylococcus aureus* 25923. **** *p* < 0.0001.

**Figure 2 pharmaceuticals-16-01112-f002:**
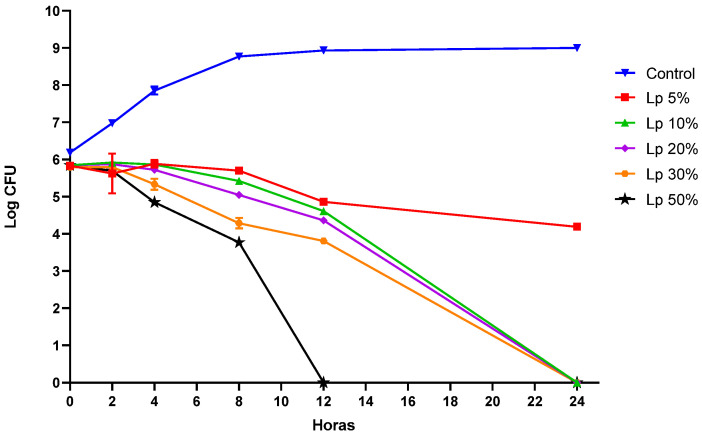
Time-kill-curve for *Staphylococcus aureus* ATCC 25923 co-incubated with *Lactiplantibacillus plantarum* ATCC 14917. Control: growth of *S. aureus* ATCC 25923 in the absence of *L. plantarum* ATCC 14917; Lp 5%: growth of *S. aureus* ATCC 25923 in the presence of *L. plantarum* ATCC 14917 at 5%; Lp 10%: growth of *S. aureus* ATCC 25923 in the presence of *L. plantarum* ATCC 14917 at 10%; Lp 20%: growth of *S. aureus* ATCC 25923 in the presence of *L. plantarum* ATCC 14917 at 20%; Lp 30%: growth of *S. aureus* ATCC 25923 in the presence of *L. plantarum* ATCC 14917 at 30%; Lp 50%: growth of *S. aureus* ATCC 25923 in the presence of *L. plantarum* ATCC 14917 at 50%.

**Figure 3 pharmaceuticals-16-01112-f003:**
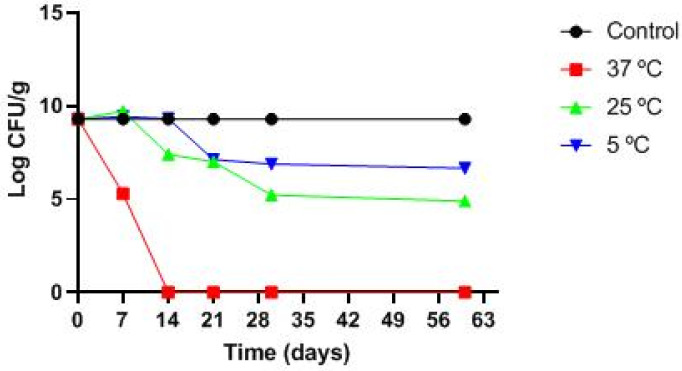
Viability of *Lactiplantibacillus plantarum* ATCC14917 in alginate gel during storage at different temperatures.

**Table 1 pharmaceuticals-16-01112-t001:** Antimicrobial activity of *Lactiplantibacillus plantarum* ATCC 14917.

Percentages Analyzed	Cells	Sonicated	Supernatant pH 5.0	Supernatant pH 7.0
5%	+*	+	+	+
10%	+*	+	+	+
20%	+*	+	−	+
30%	−	+	−	+
40%	−	+	−	+
50%	−	+	−	+

(+*): percentage that showed a decrease in the pathogenic agent; (−): percentage that presented inhibition of the pathogenic agent; (+): percentage that did not show inhibition of the pathogenic agent.

**Table 2 pharmaceuticals-16-01112-t002:** Anti–*S. aureus* effects of probiotics and postbiotics gels.

Percentages Added to the Gel	Inhibition Zones (mm ± SD)		
	Gel with Cells	Gel with Sonicated Cells	Gel with Supernatant (pH 5.0)
5%	0 ± 0 ^a,1^	0 ± 0 ^a,1^	0 ± 0 ^a,1^
10%	4 ± 0 ^a,2^	0 ± 0 ^b,1^	0 ± 0 ^b,1^
20%	4.9 ± 0.02 ^a,3^	0 ± 0 ^b,1^	6 ± 0 ^c,2^
30%	5.9 ± 0.4 ^a,4^	0 ± 0 ^b,1^	6.9 ± 0.007 ^c,3^

In each row, different superscript letters (^a,b,c^) indicate significant differences (*p* < 0.05). In each column, different superscript numbers (^1,2,3,4^) indicate significant differences (*p* < 0.05).

**Table 3 pharmaceuticals-16-01112-t003:** Organoleptic and physical–chemical characteristics of gel containing *L. plantarum* ATCC 14917 in the preliminary stability test.

Time	Aspect	Color	Odor	pH	Density
T0	N	N	N	5.3	0.890 g/mL
T2	N	N	N	4.9	1.154 g/mL
T4	N	N	N	5.0	0.720 g/mL
T6	N	N	N	4.7	0.788 g/mL
T8	N	N	N	5.1	0.885 g/mL
T10	N	N	N	5.0	0.624 g/mL
T12	N	N	N	4.9	0.606 g/mL
T14	N	N	N	4.9	0.741 g/mL

**Table 4 pharmaceuticals-16-01112-t004:** Organoleptic and physicochemical characteristics of gel containing *L. plantarum* ATCC 14917 in the accelerated stability Test.

Temperature	Time	Aspect	Color	Odor	pH	Density	Spreadability
37 ± 2 °C	T0	N	N	N	5.3	0.890 g/mL	3957 mm
	T7	LM	LM	LM	5.0	0.714 g/mL	X
	T15	M	LM	LM	5.1	0.994 g/mL	X
	T30	M	M	LM	5.4	0.705 g/mL	X
	T60	M	M	LM	4.4	0.652 g/mL	3956 mm
25 ± 2 °C	T0	N	N	N	5.3	0.890 g/mL	3957 mm
	T7	N	N	N	4.9	0.905 g/mL	X
	T15	N	N	N	5.4	0.990 g/mL	X
	T30	N	LM	LM	5.0	0.610 g/mL	X
	T60	N	LM	LM	4.7	0.766 g/mL	3367 mm
5 ± 2 °C	T0	N	N	N	5.3	0.890 g/mL	3957 mm
	T7	N	N	N	5.0	0.931 g/mL	X
	T15	N	N	N	5.4	0.938 g/mL	X
	T30	N	N	N	5.1	0.733 g/mL	X
	T60	N	N	N	4.6	0.759 g/mL	3471 mm

**Table 5 pharmaceuticals-16-01112-t005:** Result in CFU/g of the microbiological test performed with the gel at time 0 and after 60 days of formulation.

Microorganism	37 °C (0 Day)	37 °C (60 Days)	5 °C (0 Day)	5 °C (60 Days)
Bacteria Mesophiles Total aerobics	<10^3^ CFU	<10^3^ CFU	<10^3^ CFU	<10^3^ CFU
Fungi/yeasts	<10^3^ CFU	52 × 10^2^ CFU	<10^3^ CFU	<10^3^ CFU

CFU: colony forming unit.

**Table 6 pharmaceuticals-16-01112-t006:** Parameters adopted after product handling.

Parameters	Results
Aspect	Medium viscosity gel
Color	Brown
Odor (fragrance)	Characteristic
pH	5.3
Density	0.890 g/mL
Spreadability	3957 mm

## Data Availability

Not applicable.

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
