# Peer review of "The Potential of Lactiplantibacillus plantarum ATCC 14917 in the Development of Alginate-Based Gel Formulations with Anti–Staphylococcus aureus Properties"

_pharmaceuticals, 2023, doi:10.3390/ph16081112_

Round 1

Reviewer 1 Report

1. Overall the introduction is written poorly. Sentences are difficult to understand. There are major grammatical errors and too many adjectives through out. 

2. Give brief description of the Lacticaseibacillus rhamnosus ATCC 95 10863, Lactiplantibacillus plantarum ATCC 14917 Limosilactobacillus fermentum ATCC 23271 strains. are these strains skin probiotic or what?

3. Overall writing is very erratic. Methodology is not clear, too many unnecessary details. Results are complex to understand. 

Line  16   the BAL?

Line 16  The best...avoid using such words

Line 17. antagonism inhibition?  not a proper term

Line 18  can you avoid "better"

Line 21 more intense...use scientific terms  i,e significant at P Value.

Line 22  again better

Line 30-33. very long and ambiguous sentence. Cut it into two.

Line 33. Of the skin [1]. ?

Line 38.   Of the skin [1].?

Line 136 BAL ?

Line 326 Table 1. ATCC?

Line 340.  Table 2. MIC of cells.   Where are MICs?

Figure 1. Blue colour indicate S. aureus. I think all are Staph counts. Figure not labelled properly and is not self explanatory. What about the effect of MRS media components on growth of Staph?

Line 363: challenged against cosmetic formulations containing cells?

Author Response

Dear reviewer, thank you so much for all of your contributions. They were essential to the improvement of our manuscript. The changes performed are in yellow in the updated manuscript. Following we provide a point-to-point response for your comments.

  1. Overall the introduction is written poorly. Sentences are difficult to understand. There are major grammatical errors and too many adjectives through out. 

Our response: The manuscript was deeply revised to improve the grammar quality.

  1. Overall writing is very erratic. Methodology is not clear, too many unnecessary details. Results are complex to understand. 

 Our response: The manuscript was deeply revised to improve its quality.

Review comment: Line 16 the BAL?

Our response: We have changed this term for LAB throughout the text.

Review comment: Line 16  The best...avoid using such words

Our response: We have rewritten the abstract to improve the understanding.

Review comment: Line 17. antagonism inhibition?  not a proper term

Our response: We have rewritten the abstract to improve the understanding

Review comment: Line 18  can you avoid "better"

Our response: We have rewritten the abstract to improve the understanding

Review comment: Line 21 more intense...use scientific terms  i,e significant at P Value.

Our response:  We have rewritten the abstract to improve the understanding

Review comment: Line 22  again better

Our response: We have rewritten the abstract to improve the understanding

Review comment: Line 30-33. very long and ambiguous sentence. Cut it into two.

Our response: We have rewritten the abstract to improve the understanding

Review comment: Line 33. Of the skin [1]. ?

Our response: We have corrected this mistake.

Review comment: Line 38.   Of the skin [1].?

Our response: We have corrected this mistake.

Review comment: Line 136 BAL ?

Our response: We have corrected this mistake.

Review comment: Line 326 Table 1. ATCC?

Our response: We have corrected this mistake.

Review comment: Line 340.  Table 2. MIC of cells.   Where are MICs?

Our response: We have corrected this mistake.

Review comment: Figure 1. Blue colour indicate S. aureus. I think all are Staph counts. Figure not labelled properly and is not self explanatory. What about the effect of MRS media components on growth of Staph?

Our response: We have improved the figure legend.

Review comment: Line 363: challenged against cosmetic formulations containing cells?

Our response: We have corrected this mistake.

Reviewer 2 Report

This article evaluated antibacterial activity of different lactic acid bacterial strains towards S. aureus that is considered as one of the main causes of skin pathogens, by means of antagonism test-spot overlay method; then according to the obtained results, L. plantarum was selected as the effective probiotic strain. Moreover, cosmetic gel formulation was prepared based on pure sodium alginate gel and loaded with probiotic strain. After assessment of cosmetic gel stability at different temperatures and pHs, their antibacterial activity was evaluated towards bacterial, fungal and yeast strains.

This article can be published after some major revision:

1.       What is the novelty point of this work?

2.       Since antibacterial activity of probiotics and cosmetic gel were performed by measuring inhibition halo and counting CFU, it would be helpful to add some images of CFU plates or inhibition halos.

3.       There are some typos, for example in line 16 it should be LAB not BAL.

4.       English writing is weak and there are many typos and grammatical errors as following:

1.       In general, “in vitro” is written in italic.

2.       What are the numbers in keywords?

3.       There are some incomplete sentences, for example “. Of the skin.” In line 33.

4.       English writing should be improved. Some parts of the text are not understandable such as line 38 “….. to the constitution of the microbiome others”. In addition, there are grammatical errors.

5.       There are some long and unclear sentences, for example lines 52 to 58 or 71-73.

6.       For the first citation of bacterial strain, it is required to be written in full name. Later, it can be cited in abbreviations for example: S. aureus.

7.       The first citation of ATCC should be in full and complete name.

8.       Please unify the name of bacterial strain, they should be in italic.

9.       In line 107, the number of bacteria were in CFU/ml on the McFarland scale while in line 112 they were written as 0.5. Please unify them either in CFU/ml or McFarland scale.

10.   Line 115: the formula is not clear if only Dspoy divided into 2 or (Disnib – Dspoy) / 2. Should the formula be equal to something?

11.   The companies’ name of instruments must be followed with the name of city and country, for example Tryptic Soy Agar media (TSA, company name, city, country).

12.   Line 138: Ph or pH?

13.   Please unify the SI unit of volume and it was suggested that mL and µL will be used.

14.   Lines 181 to 185 are not clear.

15.   The number of tables is not correct. There are two “Table 1” and one of them was not mentioned in the text.

16.   Font size of the lines 220 – 223 changed.

17.   Generally, the section of statistical analysis presents as the last part of materials and methods; which p value was considered as statistically significant differences? It should be mentioned p < 0.05 or p < 0.01.

18.   Section 2.10.1.1: the authors mentioned several times that this experiment was performed in three replicates.

19.   Section 2.10.1.2: temperature 10 ± 2 °C is not freezer temperature.

20.   No references were cited in section 2.9 and whole section 2.10.

21.   Section 2.11: which fungal and yeast strains were used for microbiological tests.

22.   Quality of figure 1 is low.

23.   Table 3 is not understandable.

Author Response

Dear reviewer, thank you so much for all of your contributions. They were essential to the improvement of our manuscript. The changes performed are in yellow in the updated manuscript. Following we provide a point-to-point response for your comments.

This article evaluated antibacterial activity of different lactic acid bacterial strains towards S. aureus that is considered as one of the main causes of skin pathogens, by means of antagonism test-spot overlay method; then according to the obtained results, L. plantarum was selected as the effective probiotic strain. Moreover, cosmetic gel formulation was prepared based on pure sodium alginate gel and loaded with probiotic strain. After assessment of cosmetic gel stability at different temperatures and pHs, their antibacterial activity was evaluated towards bacterial, fungal and yeast strains.

This article can be published after some major revision:

  1. What is the novelty point of this work?

Our response: We have change this term for LAB throughout the text.

  1. There are some typos, for example in line 16 it should be LAB not BAL.

Our response: We have changed this term for LAB throughout the text.

  1. English writing is weak and there are many typos and grammatical errors as following:

Our response: We have rewritten the text to improve the understanding

  1. In general, “in vitro” is written in italic.

Our response: We have corrected this throughout the text.

  1. What are the numbers in keywords?

Our response: We have corrected this.

  1. There are some incomplete sentences, for example “. Of the skin.” In line 33.

Our response: We have corrected these sentences.

  1. English writing should be improved. Some parts of the text are not understandable such as line 38 “….. to the constitution of the microbiome others”. In addition, there are grammatical errors.

Our response: We have rewritten the manuscript to improve the understanding

  1. There are some long and unclear sentences, for example lines 52 to 58 or 71-73.

Our response: We have rewritten the manuscript to improve the understanding

  1. For the first citation of bacterial strain, it is required to be written in full name. Later, it can be cited in abbreviations for example: S. aureus.

Our response: We have corrected this throughout the text.

  1. The first citation of ATCC should be in full and complete name.

Our response: We have corrected this throughout the text.

  1. Please unify the name of bacterial strain, they should be in italic.

Our response: We have corrected this throughout the text.

  1. In line 107, the number of bacteria were in CFU/ml on the McFarland scale while in line 112 they were written as 0.5. Please unify them either in CFU/ml or McFarland scale.

Our response: We have improved the method description

  1. Line 115: the formula is not clear if only Dspoy divided into 2 or (Disnib – Dspoy) / 2. Should the formula be equal to something?

Our response: We have improved the method description

  1. Line 138: Ph or pH?

Our response: We have corrected this throughout the text.

  1. Please unify the SI unit of volume and it was suggested that mL and µL will be used.

Our response: We have corrected this throughout the text.

  1. Lines 181 to 185 are not clear.

Our response: We have corrected this sentence.

  1. The number of tables is not correct. There are two “Table 1” and one of them was not mentioned in the text.

Our response: We have corrected this

  1. Font size of the lines 220 – 223 changed.

Our response: We have corrected this

  1. Generally, the section of statistical analysis presents as the last part of materials and methods; which p value was considered as statistically significant differences? It should be mentioned p < 0.05 or p < 0.01.

Our response: we have adjusted this.

  1. Section 2.10.1.1: the authors mentioned several times that this experiment was performed in three replicates.

Our response: we have adjusted this.

  1. Section 2.10.1.2: temperature 10 ± 2 °C is not freezer temperature.

Our response: we have adjusted this.

  1. No references were cited in section 2.9 and whole section 2.10.

Our response: The physicochemical stability tests of the formulations were carried out by the specifications of the Quality Control and Stability Guide for Cosmetic Products and Brazilian Pharmacopoeia 2nd edition [23].

  1. Section 2.11: which fungal and yeast strains were used for microbiological tests.

Our response: In this test we searched the presence of yeast and fungal strains as contaminants.

  1. Quality of figure 1 is low.

Our response: We have adjusted this

  1. Table 3 is not understandable

Our response: We have adjusted this

Reviewer 3 Report

Comments to the Author

First of all, I would like to thank the authors for their hard work. Overall, this is a well-crafted manuscript for reading, exposition, fluency, and attention to the reader. My comments are available below.

Best regards.

Overall the introduction is well written. Expression fluency is also good. Only small parts can be corrected

1.      The summary could be written a little more effectively. Especially the results.

2.      Line 133: All microorganisms should be written in italics.

3.      Line 181: “t1hj2remf2hg235-çhe” what does it mean?

4.      Although it is a generally suitable article, it is quite remarkable that there are too many authors (Author Contribution). Too many people for this article.

Author Response

Dear reviewer, thank you so much for all of your contributions. They were essential to the improvement of our manuscript. The changes performed are in yellow in the updated manuscript. Following we provide a point-to-point response for your comments.

First of all, I would like to thank the authors for their hard work. Overall, this is a well-crafted manuscript for reading, exposition, fluency, and attention to the reader. My comments are available below.

Best regards.

Overall the introduction is well written. Expression fluency is also good. Only small parts can be corrected

  1. The summary could be written a little more effectively. Especially the results.

Our response: We have adjusted the abstract

  1. Line 133:All microorganisms should be written in italics.

Our response: We have adjusted this

  1. Line 181:“t1hj2remf2hg235-çhe” what does it mean?

Our response: We have adjusted this

Round 2

Reviewer 1 Report

Manuscript is improved now